# The Application of an Eco-Friendly Synthetic Polymer as a Sandy Soil Stabilizer

**DOI:** 10.3390/polym15244626

**Published:** 2023-12-06

**Authors:** Nathália Freitas Boaventura, Thiago Freitas de Porfírio Sousa, Michéle Dal Toé Casagrande

**Affiliations:** 1Department of Civil and Environmental Engineering, University of Brasilia, Brasilia 70910-900, Brazil; mdtcasagrande@unb.br; 2Companhia de Desenvolvimento dos Vales do São Francisco e do Parnaíba (Codevasf), Montes Claros 39400-292, Brazil

**Keywords:** composites, synthetic polymer, reinforcement, new geomaterials, soil stabilizer

## Abstract

The conducted investigation encompassed the comprehensive integration of mechanical, environmental, chemical, and microstructural evaluations of a composite amalgamating sandy soil and a synthetic polymer at two distinct concentrations (2.5% and 5%) across multiple curing intervals (0, 1, 2, 4, 7, 15, 30, and 45 days). The studied soil originates from an environmentally significant protected area in Brazil. The implementation of mechanisms for soil improvement in the region must adhere to technical criteria without causing environmental harm. Direct shear testing was conducted, permeability was assessed, and microstructure analysis and XRD and XRF/EDX studies of both the soil and composites were conducted. It was observed that longer curing times yielded improved results in shear stress, friction angle, and cohesive intercept, with SP_5% exhibiting the highest values compared with the soil and SP_2.5%. Adding the polymeric solution to the soil contributed to cementation and cohesion gains in the substrate. Through microstructural characterization, the polymer’s role as a cementing agent for the grains is evident, forming a film on the grains and binding them together. Based on the analyses and studies conducted in the research, it can be concluded that there is technical feasibility for applying the polymeric solution at both dosages in geotechnical projects.

## 1. Introduction

Soil is a common element in geotechnical works, exhibiting significant variation and stratification among regions, often necessitating stabilization and, consequently, improvement of the underlying substrate [1]. Many geotechnical projects aim at soil stabilization, wherein the success of soil enhancement procedures is evaluated, striving to promote soil stability, strength, resistance to erosion, and economic viability. This holds for stabilization, improvement, or reinforcement of diverse geotechnical structures [1,2,3,4].

When considering the use of polymers in soil stabilization, they can be categorized into three major groups: geopolymers, biopolymers, and synthetic polymers [5]. Geopolymers are inorganic substances of aluminosilicate with combinations of other elements [4,6,7]. Biopolymers are produced based on bacteria biomass and are common in nature [8,9,10]. Finally, synthetic polymers, created in laboratories, are based on main and branched-chain structures [11,12,13].

Synthetic polymers, being man-made, can be manipulated to acquire desired characteristics. Consequently, many synthetic polymers are being developed to be less harmful to the environment (from production to use), more durable, and economically viable for soil stabilization [11,12,13].

The prioritization of environmentally friendly materials to enhance soil quality has become a widespread trend, driven by increasing environmental awareness. With growing concern about environmental impact, various investigations are being conducted into new materials deemed “eco-friendly”. Using polymers as modifiers of soil structures appears promising, improving the microstructure of mixtures and enhancing the durability of composites [4,14,15]

“Green Chemistry” [16] provides a framework for creating a more environmentally friendly chemical, process, or product. The non-use of conventionally employed materials (such as cement and lime) in favor of synthetic polymers is based on several principles of green chemistry, resulting in significant environmental benefits. The main environmental concern does not lie in the use of synthetic polymers per se, as evidence suggests a reduced environmental toxicological impact [12]. The focus is on the production of these polymers, associated with chemicals derived from potentially harmful substances [17], although they are considerably less harmful than traditionally used materials [11].

When discussing conventionally used materials, one example is cement. The production of cement, which involves heating the clinker to 1450 °C in a kiln, results in the release of a considerable amount of carbon dioxide (CO_2_) [11,18]. In response to this negative impact, researchers are developing environmentally friendly alternatives to enhance the properties of weak soil [11].

Despite the relevance of synthetic polymers, especially water-soluble ones, to society at large [12], there is a small environmental risk when compared with biopolymers [17]. Many of these polymers are derived from petroleum, raising environmental concerns [19]. The principles of “Green Chemistry” should be considered when studying materials intended for introduction into nature [20].

The study by Al-Khanbashi and El-Gamal [21] on acrylic styrene emulsion in sandy soil in the Al Ain desert, United Arab Emirates, addressed hydraulic conductivity and unconfined compression resistance for different polymer solution dosages and two compaction methodologies. Over the curing period, the sandy soil solidified, showing significant improvements in mechanical and hydraulic properties with all percentages of added solution, compared with untreated soil.

Garcia et al. [22] observed considerable increases in cementation forces, traction, and shear strength of sandy soil with polymer addition, compared with pure sand and another sand with artificial cementation.

Okonta [23] evaluated the application of acrylic polymer solution at different curing times, percentages, and temperatures for soil characteristic of the South Africa region, noting a reduction in voids with increasing polymer solution percentage. Additionally, it was observed that curing at higher temperatures led to an increase in unconfined compression strength compared with curing at lower temperatures. The soil used had predominantly sandy characteristics, and the addition of the polymer solution proved effective in generating cementation between grains.

Barreto et al. [1] observed improvements in shear strength parameters of samples added with synthetic copolymer for different curing times and dosages compared with pure sand. Luan et al. [24] applied a vinyl acetate-ethylene synthetic polymer in clayey soil at different curing times. After the 14th day of curing, a considerable increase in strength was observed, forming a dense structure, and the longer the curing time, the greater the strength and density. These results support the addition of a polymer as a soil stabilizer.

In Brazil, there are numerous environmental protection areas where sustainable development should be a priority. An example is the Environmental Protection Areas (APAs), which aim to reconcile nature conservation with the sustainable use of environmental resources. One such APA of paramount importance to the entire Brazilian ecosystem is the APA-Jalapão, which includes Conservation Units (UC) that require sustainable use of all resources. The area combines sustainable use of environmental resources with tourists visiting to appreciate the natural beauty of the location.

Therefore, using synthetic polymers in the APA-Jalapão region for soil improvement to facilitate access for both the local population and sustainable tourism may be a possibility to mitigate the environmental impact of necessary works. This approach could lead to improved access to the location, enhanced soil resistance parameters, reduced maintenance, and increased safety and comfort for users, consequently fostering increased tourism activity and economic growth.

This study aims to investigate the utilization of a synthetic polymer, styrene acrylic copolymer, as a stabilizing agent for the soil in the APA-Jalapão region. The study involves the addition of the copolymer at two different dosages, and evaluating direct shear strength and soil parameters (friction angle, cohesive intercept, and permeability). Additionally, the microstructure of the soil after improvement is assessed.

## 2. Materials and Methods

### 2.1. Soil and Composites 2.5% and 5%

The materials employed in this study comprised sandy soil sourced from the northern region of Brazil and a polymeric solution (Figure 1).

The supplier provided the polymer stabilizer used in this research. It consists of an organic acrylic-styrene copolymer obtained randomly and presented as an aqueous emulsion of anionic character. The polymer has a pH of 8.0–9.0, a density of 0.98–1.04 g/cm^3^, a viscosity of 3000–10,000 centipoise (cP), and is completely soluble in water. CHN Elemental Analysis showed that the polymer comprises 69.03% carbon, 7.01% hydrogen, and 0.52% nitrogen. The product is frequently used for pavement sealing and as a soil stabilizer [2,3,6].

The decision to use this particular polymer was based on its prior use in the study region (APA-Jalapão); however, in-depth analyses regarding its specific application in the local soil had not yet been conducted.

The chemical identification by the Chemical Abstracts Service (CAS) system is 25035-69-2, with the molecular formula C_16_H_26_O_6,_ and a molecular weight of 314.38 u. Its chemical name is Methacrylic acid, a polymer with butyl acrylate and methyl methacrylate, with acrylic-styrene copolymer being a synonym. The Chemical Book (CBnumber) assigns the number CB5958971 to this polymer. Figure 2 depicts the organic function of the formulation.

For the physical characterization of the soil, the granulometry was determined according to standard NBR 7181 [25], similar to ASTM D7928-21e1 [26]. In addition, the specific gravity of the soil was measured using a PENTAPYC 5200e pycnometer, and ASTM D5550-14 “Standard Test Method for Specific Gravity of Soil Solids by Gas Pycnometer” [27].

The curing times evaluated were 0, 1, 2, 4, 7, 15, 30, and 45 days. The curing days were counted after the specimen was made. The specimens were cured by exposure to air since polymer flocculation occurs through exposure to atmospheric oxygen.

### 2.2. Proctor Compaction Tests

Proctor Compaction Tests with normal compaction energy were performed for the soil and the composites, where the polymeric solution was added [28,29]. Two dosages were made for the study. The first solution contained 25% polymer and 75% water, i.e., a ratio of 1:4 of the total volume. The second solution contained 50% polymer and 50% water, i.e., a ratio of 1:2 of the total volume. The amount of polymer solution inserted (optimum moisture content) was calculated according to the mass of the soil. Therefore, the soil-polymer composite had 2.5% and 5% polymer by soil mass, respectively.

The polymer composition in the dilution, established at 25%, was chosen due to its previous application in regional projects. Meanwhile, the 50% concentration in the dilution was adopted to perform an extrapolative analysis concerning the existing dosage, thus resulting in the double dilution of the polymer.

### 2.3. Direct Shear Test

The direct shear test was conducted to determine the soil strength parameters (cohesion and friction angle) from establishing the Mohr-Coulomb strength envelope [30]. First, compaction was carried out with normal energy in a large cylinder, representing possible grain breakage and the interaction between the soil elements and the solution. Subsequently, the samples were thinned using a mold, which had the dimensions of the small shear box (60 mm × 60 mm × 25 mm). Normal stresses of 50 kPa, 100 kPa, 200 kPa, and 400 kPa were applied to obtain the strength envelope.

Due to its granular nature, the consolidation phase was remarkably swift. The shear rate was determined by the consolidation phase and set at 0.5 mm/min.

### 2.4. Mineralogical Characterization

Mineralogical characterization tests were carried out to identify the minerals present in the structure. The soil and composite samples were analyzed using X-ray diffraction (XRD) and X-ray fluorescence spectrometry (XRF/EDX) tests. Microstructural characterization was carried out by analyzing Optical Microscopy (OM) and Scanning Electron Microscopy (SEM) images. Tests were also carried out on the pure samples and the composites in the test specimens.

Scanning Electron Microscopes (SEM) employ electron beams as opposed to photons used in conventional optical microscopes. Samples to be analyzed have a reduced diameter and are subjected to vacuum conditions. SEM swiftly provides information regarding the morphology and identification of chemical elements present in a sample, standing out as one of the most versatile instruments for analyzing microstructural characteristics of solid objects. An additional advantage lies in the three-dimensional representation of images obtained from the samples [31].

The Energy Dispersive System (EDS) is coupled to the equipment used for acquiring SEM images, serving as an analytical technique for the chemical characterization of samples, enabling qualitative and semi-quantitative determination of their composition. In this method, an electron beam is focused on the sample in SEM, interacting with the atoms present and releasing energy in the form of X-rays. Following the qualitative analysis of the soil grain microstructure, a semi-quantitative determination of the chemical composition was performed [31].

Preceding the microscopy phase, the soil samples underwent a high-vacuum evaporation metalization process. This procedure results in the deposition of an ultra-thin layer of electrically conductive material on the samples; in this case, gold was used.

For obtaining Scanning Electron Microscopy (SEM) images and conducting X-ray fluorescence spectrometry (XRF/EDX) tests, the equipment used was a JEOL JSM 7100F Scanning Electron Microscope by JEOL BV (Zaventem, Belgium) with an acceleration voltage of 30 KV, a resolution of 3nm, and magnification up to 300,000× with an EDS X-ray microanalysis system.

Optical microscopes, instruments that use light refraction in lenses to magnify and visualize elements under analysis, allow the observation of structures that escape the naked eye’s perception. The instrument used for Optical Microscopy (OM) analyses was a Celestron LC Professional Digital Microscope model 44345 by Celestron (Sao Paulo, Brazil), equipped with automatic magnification lenses ranging from 4× to 40×, along with a 90 mm LCD monitor with 4× digital zoom and a high-definition TFT display.

X-ray Diffraction (XRD) analyses were conducted using the Rigaku Ultima IV X-ray Diffractometer by Malvern Panalytical (Malvern, United Kingdom). XRD enables the identification of minerals by characterizing their crystalline structures. The process was carried out using material passing through a #200 sieve, both for sandy soil and the compounds under analysis.

## 3. Results and Analysis

### 3.1. Soil

The soil has a specific gravity of 2.7 g/cm^3^ and is classified as a non-plastic soil (NP). The soil’s grain size distribution curve (Figure 3) following the HRB classification adopted by AASHTO corresponds to soil type A-3. Utilizing the Soil Unified Classification System (SUCS), the studied soil is classified as SP—poorly graded sand.

### 3.2. Proctor Compaction Tests

The compaction curves for both the soil and the composites are depicted in Figure 4. There is a noticeable alteration in the behavior of the soil compaction curve with the addition of the polymer. An increase in polymer dosage is associated with a tendency for the compaction curve to become more concave, thereby resembling the anticipated compaction curves for soils with clay-like behavior.

The pair of optimum moisture content and maximum dry density were utilized for the tests. However, to achieve this pair, 1.5% of the water-polymer solution must be added during specimen preparation.

The constant head permeability test was conducted on the soil, yielding a coefficient of permeability value of 5 × 10^−3^ cm/s. This value characterizes a soil with high permeability, typical of fine sandy soils.

### 3.3. 2.5% Polymer Composite

Figure 5 displays the shear stress versus horizontal displacement curves for the soil with added water (optimum moisture content for normal energy) for the application of normal stresses of 50 kPa, 100 kPa, 200 kPa, and 400 kPa.

In Figure 6a–h, the shear stress versus horizontal displacement curves are presented for each curing time and for the application of normal stresses with 2.5% polymer.

The influence of curing time on the composites is evident from the shear stress curves. Tests conducted with 0 and 1 day of curing exhibited curves with a shape similar to that for sandy soil, without a distinct peak and with asymptotic behavior. As the curing time increased, the peaks of strength became noticeable, with the peak becoming more evident and pronounced over the days.

Figure 7 displays the failure envelopes of the composites with different curing times for 2.5% polymer. Table 1 presents the values of effective friction angle and cohesive intercept for each curing time.

Due to the soil used in the study having less than 12% fines content, the overall behavior of the soil is governed by the friction angle. Thus, by adding the polymer, an increase in the cohesive intercept is expected, as the additive works in cementing the grains.

In Figure 6a,b, a reduction in load capacity is observed compared with the soil curves (Figure 5). The soil, when mixed with the polymeric solution, initially exhibits a pasty behavior, reducing both grain interlocking and cementation, consequently leading to a decrease in load capacity.

Analyzing the results of sample SP_2.5%_1 (Table 1), an increase in the cohesive intercept is observed, indicating hardening of the outer layer of the specimen. However, internally, it was still moist due to curing occurring with direct exposure to air. For this reason, a lower friction angle is seen compared with pure soil. In Figure 8a, the rupture of one of the specimens after the direct shear test, with 1 day of curing, can be observed.

From the second day of the curing process onwards, a notable increase in the analyzed strength parameters is observed. This phenomenon can be attributed to the increase in internal stiffness of the specimens after a 48-h period. This increase in stiffness indicates the progress of curing to the deeper layers of the specimen, accompanied by an intensification in the adherence of the polymer to soil particles, thus promoting greater grain cementation. Figure 8b illustrates the rupture of a specimen with two days of curing, showing evident drying through the observed color variation in the image. However, it is important to note the presence of loose grains inside the specimen, indicating that complete grain cementation did not occur in the sample.

There was a notable increase of over 200% in the value of the cohesive intercept when comparing SP_2.5%_4 with SP_2.5%_2. This result unequivocally confirms that the curing process plays a fundamental role in enhancing soil properties. It is important to note that the friction angle also recorded an increase, although remaining within the limits observed in the literature for sandy soils.

In Figure 9, there is visual evidence of the curing process, where no moisture is observed in the internal portion of the specimens. Regarding longer curing periods, an increase in the value of the cohesive intercept was noted, while the friction angle showed few variations. These observations are in line with the reduction in the presence of loose particles as the curing time increases.

In the external portion of all specimens, which is directly exposed to the environment (direct contact with air), greater stiffness was observed, along with a notable difference in color compared with the internal portion. This phenomenon can be attributed to the fact that the curing process occurs from the outside in. It is believed that the increase in the friction angle value is attributed to the increase in grain cohesion over time, given the little variation in this parameter. With the increase in curing time, the specimens had a similar rupture, corroborating the found results (Figure 9).

### 3.4. 5% Polymer Composite

In Figure 10, the shear stress versus horizontal displacement curves are presented for the soil with the addition of water (optimal moisture content for normal energy) for the application of normal stresses of 50 kPa, 100 kPa, 200 kPa, and 400 kPa. In Figure 11a–h, the shear stress versus horizontal deformation curves are presented for each curing time and for the normal stresses for 5% polymer.

It is observed that the behavior of the curves is similar to that of composites with a lower amount of polymer. However, after reaching the peak at the stress of 400 kPa for 7 days of curing, there is a sharp decline, and the same pattern is observed for all curves at 15, 30, and 45 days. Due to machine limitations, it was not possible to perform tests with the normal stress of 400 kPa for 15, 30, and 45 days of curing. Nevertheless, this does not pose an issue, as it is possible to obtain rupture envelopes and conduct a detailed analysis.

Figure 12 displays the rupture envelopes of the composites with different curing times for 5% polymer. Additionally, Table 2 presents the values of effective friction angle and cohesive intercept for each curing time.

The samples with addition of the uncured polymeric solution, SP_5%_0, exhibited inferior behavior in the obtained parameters and lower strength values compared with both the soil and SP_2.5%_0. As the polymer content in the solution increased, greater grain lubrication was observed, resulting in a lower friction angle and the absence of cohesive intercept. In Figure 13a, higher moisture content in the specimen is evident, along with its state after the test. However, superior behavior for both parameters (cohesive intercept and friction angle) is already observed with just 1 day of composite curing. Additionally, with 1 day of curing, the specimen, despite still being moist, shows a perception of cohesion between the grains (Figure 13b).

With two days of curing, there is noticeable improvement in both parameters, with an increase of approximately 3000% in the cohesive intercept compared with the soil without polymer. Improvement in the behavior of SP_5%_2 is evident, despite the presence of some internal moisture (Figure 13c), both in terms of parameters and strength values, which were only observed in the 2.5% dosage from the 4th day of curing. This moisture is indicated by the central part of the sample having a different color from the edges and by the detachment of the interior after the test.

In the SP_5%_4 samples, there was an increase in the cohesive intercept value of approximately to 40% when compared with the soil sample. When comparing the friction angle with the result obtained for the SP_2.5%_2 samples, there is an increase of approximately 30%. However, the cohesive intercept of SP_2.5%_4 is higher than the value found for the composite with a higher polymer content. This is attributed to the increased polymer in the curing solution, slowing down the curing process inside the sample. However, there was less detachment of grains in the center of the sample after the test. Another point to consider is that for composites with a lower polymer content, the values stabilize from the 4th day of curing, which does not occur for samples with a higher polymer content, where the values continue to increase.

With 7 days of curing, there was a clear increase in both parameters. The cohesive intercept value for this curing time is practically double the value found for SP_2.5%_45. Analyzing the friction angle value found for the composite with this dosage and from the 7th day of curing, the results are similar to typical intervals of friction angle values for rocks, such as granite (45–60°), dolerite (55–60°), sandstone (35–50°), marble (35–50°), and basalt (50–55°).

On the 15th day of curing, there is an increase in the cohesive intercept value of approximately 70% compared with SP_5_7, and of 12800% when compared with the soil. It is noticeable that after the 15th day, there is no significant increase in either cohesive intercept or friction angle values. Figure 14a shows the rupture of a specimen with 15 days of curing, and Figure 14b shows the rupture with 45 days of curing.

Figure 15 shows the stress vs. horizontal displacement curves for both composites with 45 days of curing and for the soil.

There is a clear increase in stiffness, peak values, and residual stress values with the addition of polymer. It is evident that SP_5% is more resistant to shear than SP_2.5% and the untreated soil. In other words, the addition of the polymeric solution is advantageous, and the higher the polymer content, the greater the shear resistance of the soil for all studied confining stresses.

Figure 16 depicts all the curves for confining stresses of 50 kPa and 200 kPa for the SP_2.5% composite. Meanwhile, Figure 17 illustrates all the curves for confining stresses of 50 kPa and 200 kPa for the SP_5% composite.

For both composites, as the confining stress increases, the shear stress also increases. However, the increase in confining stress results in curves with a more pronounced peak, slightly altering the curve’s behavior. After rupture, for the curves with 200 kPa, the values of residual stresses are closer to the soil value, which is less evident for the 50 kPa values.

In Figure 18, the correlation between cohesive intercept and friction angle of the composites and the corresponding curing time is presented.

Regarding the friction angle, an increase in value was observed for the SP_2.5% dosage on the second day, followed by a decrease on the fourth day, and on the seventh day, the value stabilizes. However, it is essential to note that this increase is not truly significant, as there was a stabilization in the friction angle values for this dosage. Concerning the cohesive intercept, a significant increase was observed until the fourth day of curing, followed by stabilization on the seventh day for the SP_2.5% dosage.

For the SP_5% dosage, the friction angle showed a significant increase until the fifteenth day of curing, followed by stabilization in the subsequent days. The cohesive intercept values for this dosage also followed the same trend, stabilizing from the fifteenth day onward.

The variable load permeability coefficient test [24] was conducted for both SP_2.5% and SP_5%, and the obtained value for both was 2 × 10^−4^ cm/s. When compared with untreated soil, a significant reduction in permeability is evident with the incorporation of the polymer. This indicates reduced voids between grains, approaching the behavior of a soil with low permeability.

The polymer at both concentrations proves to be environmentally benign. This conclusion is drawn from the chemical analysis conducted on the leachate at both dosages. Considering the concentration limits set by the Brazilian National Environment Council [32,33] and the waste disposal limits of NBR 10004/2004 [29], all minimum allowable values (MAV) for chemical elements are below the established thresholds. Table 3 specifies the evaluated elements and the concentrations found.

Assessing the leachate data in light of CONAMA Resolution 420 [32], which, in Annex II, establishes maximum allowable and guidance values for soil and groundwater, the maximum permissible value (MPV) for dissolved copper in groundwater is 2 mg/L, and both samples recorded values below this threshold. Another element described in the standard is aluminum, with an MPV of 3.5 mg/L; the values for both composites were below the established norm. The last element mentioned in the resolution is manganese, with an MPV of 0.4 mg/L, and there is coherence with the values found for both composites. It should be noted that the manganese MPV is related to health risks associated with its consumption, not direct contamination from natural elements.

Treating the leachate as an effluent from a polluting source involves using CONAMA Resolution 430 [33], Section II, Article 16, which determines the conditions and standards for the discharge of effluents from any polluting source directly into the receiving body. Thus, by analyzing all elements in both leachates, it is possible, in accordance with the resolution, to meet the conditions for direct discharge into the water body.

NBR 10004 [29] establishes regulations for solid waste. In Annex F, it sets the maximum concentration values obtained in leachate tests. The MPV for all elements found complied with the limits described in the standard.

This analysis is of utmost importance as the stabilized soil studied originates from the APA-Jalapão, an environmental preservation area. Therefore, the use of the polymer will not cause environmental issues in the region, whether used in landfills and slopes or as a base or coating in pavement layers.

### 3.5. Mineralogical Characterization

#### 3.5.1. X-ray Diffraction (XRD)

A sand X-ray diffraction (XRD) analysis substantiates this by revealing a significant quantity of quartz (SiO_2_) in the soil’s mineralogical composition. Another mineral identified is goethite (FeO(OH)), commonly associated with quartz through sulfide alterations. Additionally, kaolinite (Al_2_Si_2_O_5_(OH)_4_), a clay mineral formed by weathering, is present. Figure 19 presents the XRD analysis of the sand.

Figure 20a,b presents the XRD analyses of the SP_2.5% and SP_5% composites, respectively.

The SP_2.5% composite exhibited quartz and kaolinite in its composition, along with a mineralogical structure resembling hematite (Fe_2_O_3_). Hematite is typically found as a primary mineral in various rock types. However, in this case, there has not been sufficient geological time or processes for a chemical alteration of the discovered mineral. The equipment aims to identify particles based on mineral characteristics. The observed hematite-like structure is believed to be the presence of the polymer in the soil matrix, appearing nodular and acting as a cementing agent for soil aggregates, similar to hematite.

In the analysis of the SP_5% sample, hematite is also present, along with kaolinite and quartz, as seen in the pure soil. Additionally, a small amount of gibbsite (Al(OH)_3_) appears in this analysis, associated with the clay fraction of the soil. Gibbsite is a positively charged material with low hardness. Since the soil does not have a clay fraction, this composite, with the highest polymer content, likely shows the material identified due to the increased polymer presence.

#### 3.5.2. X-ray Fluorescence Spectrometry (XRF/EDX) Tests

According to Energy Dispersive X-ray Spectroscopy (EDX), the soil’s composition includes the following elements: oxygen (61.55%), silicon (25.04%), aluminum (5.39%), carbon (4.98%), and iron (3.02%). Figure 21 presents the EDX analyses of the soil.

Figure 22a,b presents the XRF/EDX analyses of the SP_2.5% and SP_5% composites, respectively.

In the SP_2.5% and SP_5% samples, the identified elements were the same, with very close percentages. For SP_2.5%, the elements found were oxygen (48.93%), carbon (40.02%), silicon (7.34%), aluminum (1.89%), and nickel (1.82%). For SP_5%, the identified elements were oxygen (48.94%), carbon (35.94%), nickel (8.76%), silicon (4.51%), and aluminum (1.84%).

The identified elements are largely the same, except for nickel, which appeared only in the composites, attributed to the addition of the polymer. Nickel is a corrosion-resistant and mechanically resistant metal, and its presence in the composites supports the observed higher resistance. The increase in carbon concentration in the composites is attributed to the presence of the polymer, as the primary constituents of a polymer matrix are oxygen and carbon.

#### 3.5.3. Optical Microscopy (OM) and Scanning Electron Microscopy (SEM) Images

In Figure 23a–c, soil grains are observable as clear structures with well-defined borders, varying in size, predominantly featuring small, rounded, and angular grains. There is no apparent element binding the grains together, with smaller grains situated near larger ones occupying spaces. Figure 23d is magnified by 950× over the surface of a grain, revealing some irregularities on the surface, with no observable voids between the grains.

Figure 24a–d presents images obtained through optical microscopy of the SP_2.5% and SP_5% composites with a 10× magnification.

The images obtained through optical microscopy already reveal the cohesion of soil grains upon adding the polymer. It is also noticeable that there is a shinier aspect surrounding the grains, resembling glue or a film. On the external part of the composite, grains appear more united, with fewer voids than on the internal part, observed for both dosages. SP_5% seems to exhibit a greater cohesion of grains compared with SP_2.5%.

In Figure 25a–c, images obtained through scanning electron microscopy of the SP_2.5% composite are shown.

In Figure 25a, with a 15× magnification, it is possible to observe that the soil particles appear to be more united. In this type of image, depth can be seen. There is union between particles throughout the surface. In Figure 20b, with 75× magnification, the union of particles is more evident.

In Figure 25c, with 270× magnification, unions between substrate particles are observed. In certain grains, the boundaries of the grain are not visible; instead, there is a union between particles. A meniscus is also visible, connecting the particles, and at times, these menisci unite, leaving small voids. It is believed that these unions and menisci result from the addition of the polymer. Small structures are observed on the grains, representing the incorporated polymeric solution. Additionally, the image reveals cementation between the grains.

Figure 26a–c show the images obtained by scanning electron microscopy of the SP_5% composite.

In Figure 26a, at 22× magnification, the substrate particles seem even more consolidated, and cemented layers of grains are apparent across the surface. Figure 26b, at 90× magnification, shows a stronger union of particles with the same material present on the soil particles.

In Figure 26c, at 300× magnification, the unions between substrate particles are emphasized. Menisci connecting particles are observed, and in some grains, the boundaries are less apparent. The menisci seem thinner, and more concave compared with SP_2.5%. Additionally, material is evident not only on the top of the grains but also between the grains. The appearance of gibbsite and hematite in the XRD analysis for SP_5% is attributed to this presence between soil grains, acting as a cementing element. Hematite is also present in SP_2.5%.

Moreover, a thinning of the menisci and an increase in concavity are noted. This alteration is associated with the higher polymer content in the composite, justifying the observed cementation and the increase in cohesive intercept noted in the results.

## 4. Conclusions

This study proposes the utilization of a composite material for geotechnical structures. The conclusions drawn from the conducted tests are as follows:The studied soil comes from an environmentally protected area where large-scale construction is restricted, making access to the region challenging. The soil is sandy, classified as A-3 by HRB classification, non-plastic (NP), and SP by SUCS classification. Based on these characteristics and microstructural analysis, it can be affirmed that it is a soil without natural cementing elements, exhibiting granular soil behavior;Overall, the addition of polymer to the studied soil was positive for all the studied applications. The addition of the polymer solution to the soil, in two concentrations of 2.5% and 5%, provided gains in cementation and cohesion to the substrate;The compaction results showed a change in the behavior of the curves with the polymer insertion in both concentrations, along with an increase in dry density, but a slight reduction in the optimum moisture content. However, working with a polymer solution leads to a reduction in the total amount of added water;Analysis of the direct shear results for both studied concentrations showed that an increase in curing time improves the strength parameters, both cohesive intercept and friction angle;Permeability was reduced for both composites compared with pure sandy soil, attributed to the polymer solution occupying the void spaces between grains;From an environmental perspective, the leachate analysis did not reveal excessive chemical elements that could contaminate groundwater, fauna, flora, or the local population;The action of the polymer increases the rigidity of the composite upon exposure to air, as this is a common chemical reaction in any type of adhesive, where oxygen acts as a catalyst for material hardening;Microstructural characterization visibly shows the action of the polymer as a cementing agent for grains, forming a film on the grains and binding them. With the increase in curing time in mechanical tests, this grain union led to an improvement in the strength values of the composites;Through the analyses and studies conducted in the research, it can be concluded that there is technical feasibility for the application of the polymer solution, for both concentrations, in geotechnical projects. In general, the 5% content yielded better results, although for applications in embankments and slopes, the 2.5% content produced satisfactory results.

The conducted research involved the integration of mechanical, environmental, chemical, and microstructural assessments of a composite that combines soil and polymer in two different concentrations and for various curing periods. The enhanced soil underwent a refinement process, and the resulting composites were designed for use in geotechnical applications, such as slopes, embankments, and pavement. The use of polymer as a chemical stabilizing agent, as opposed to conventionally employed materials, showed promising results in various analyses, validating the effectiveness of this product as a stabilization element.

The soil comes from an environmentally protected area, meaning any infrastructure project must strictly comply with environmental regulations. However, incorporating this material, the polymer, in the studied concentrations, proved to be a technically viable and environmentally friendly solution.

## Figures and Tables

**Figure 1 polymers-15-04626-f001:**
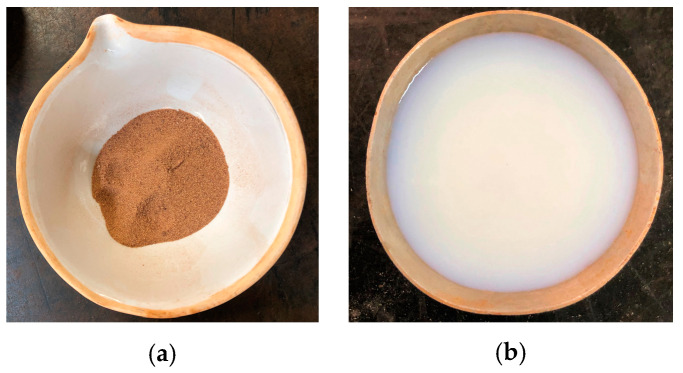
The materials employed in this study (**a**) Sandy soil sourced from the northern region of Brazil (**b**) Polymeric solution.

**Figure 2 polymers-15-04626-f002:**
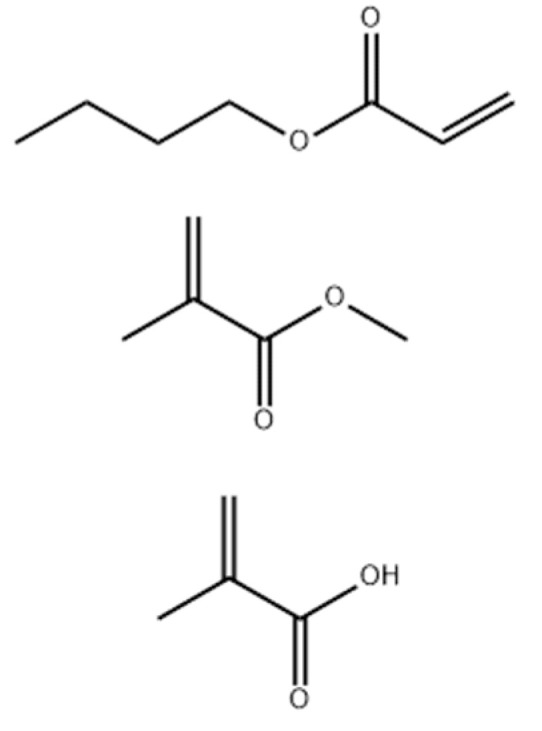
The organic function of the C_16_H_26_O_6_ formulation.

**Figure 3 polymers-15-04626-f003:**
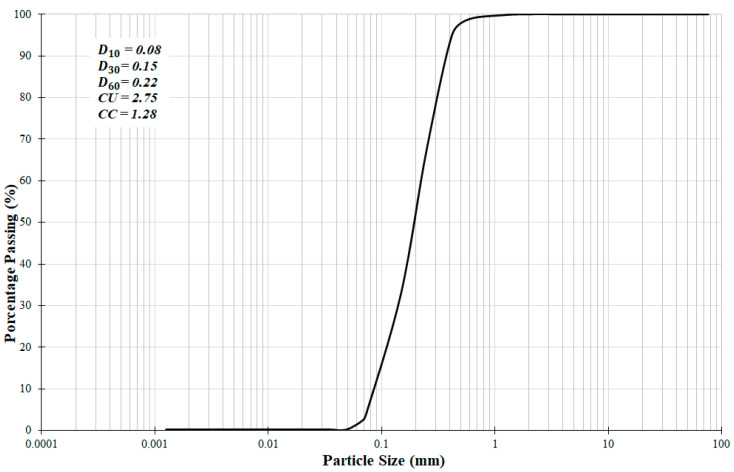
Particle grain size distribution.

**Figure 4 polymers-15-04626-f004:**
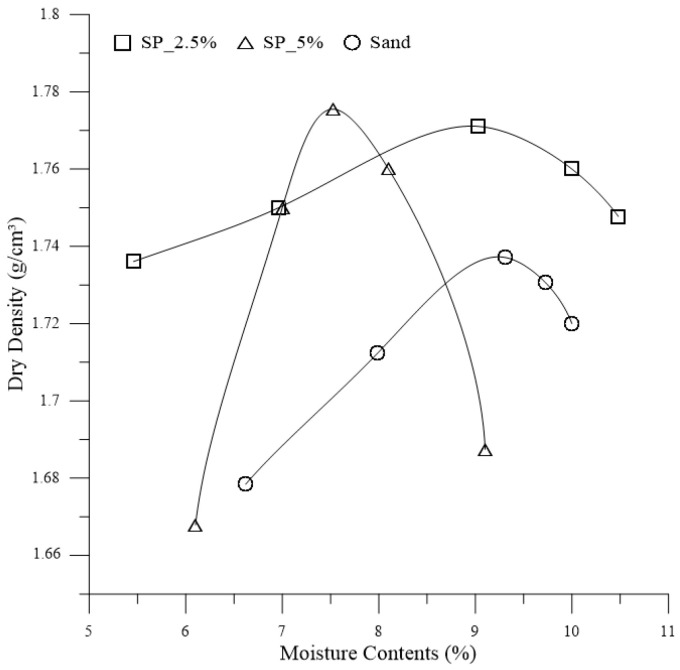
Compaction curves for soil (Sand) and composites (SP_2.5% and SP_5%).

**Figure 5 polymers-15-04626-f005:**
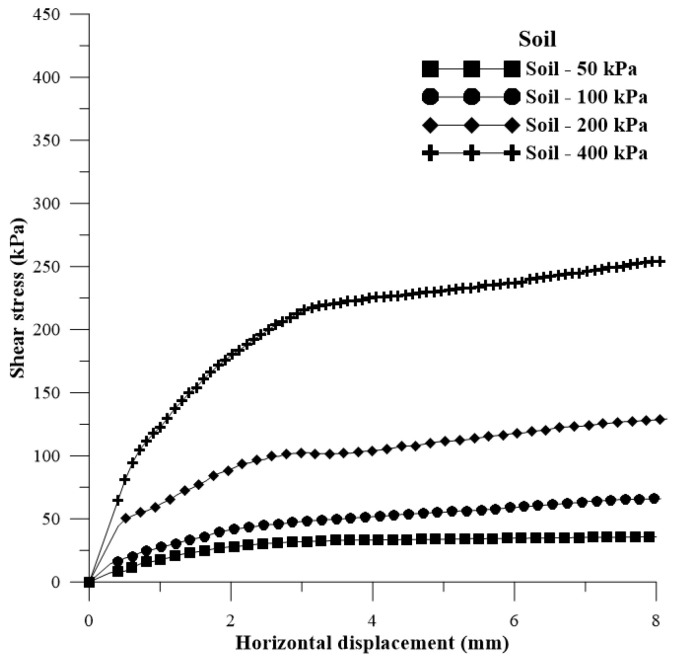
Shear stress versus horizontal displacement curves for soil + water.

**Figure 6 polymers-15-04626-f006:**
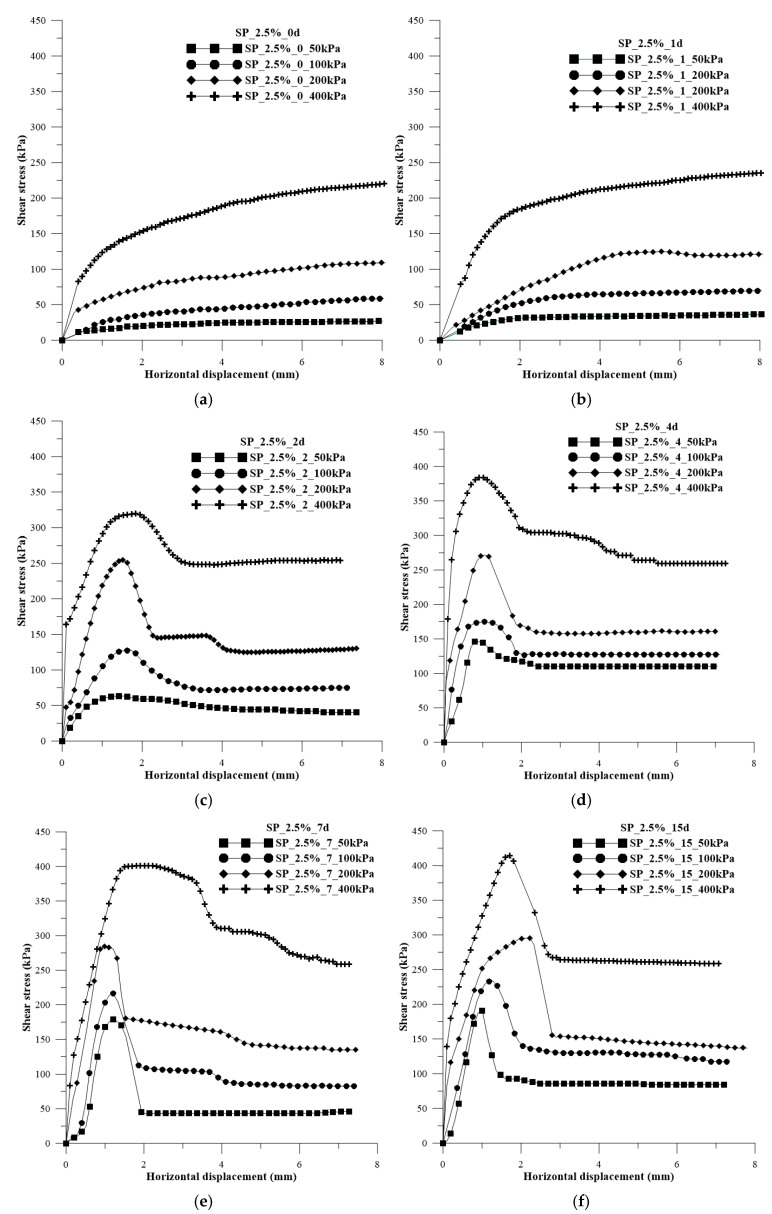
Shear stress versus horizontal strain curves for each curing time and for the application of normal stresses of 50 kPa, 100 kPa, 200 kPa, and 400 kPa with 2.5% polymer: (**a**) 0 days of curing; (**b**) 1 day of curing; (**c**) 2 days of curing; (**d**) 4 days of curing; (**e**) 7 days of curing; (**f**) 15 days of curing; (**g**) 30 days of curing; (**h**) 45 days of curing.

**Figure 7 polymers-15-04626-f007:**
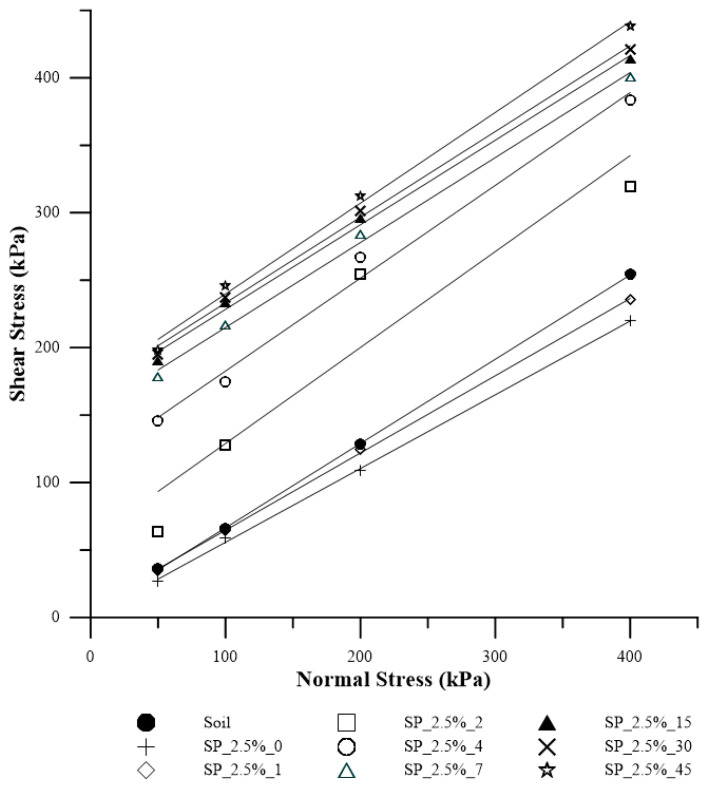
Failure envelopes for the soil and composite SP_2.5% for each studied curing time.

**Figure 8 polymers-15-04626-f008:**
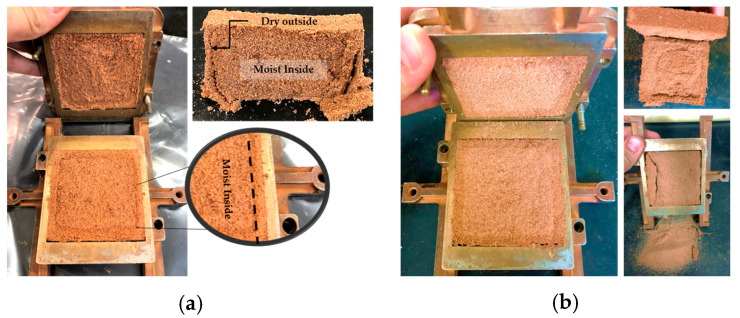
Specimen after the direct shear test: (**a**) SP_2.5%_1; (**b**) SP_2.5%_2.

**Figure 9 polymers-15-04626-f009:**
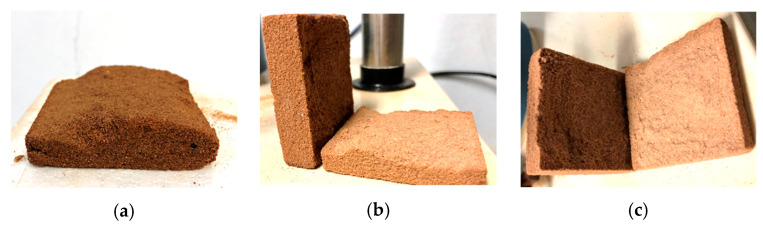
Specimens after the direct shear test: (**a**) SP_2.5%_4; (**b**) SP_2.5%_15; (**c**) SP_2.5%_45.

**Figure 10 polymers-15-04626-f010:**
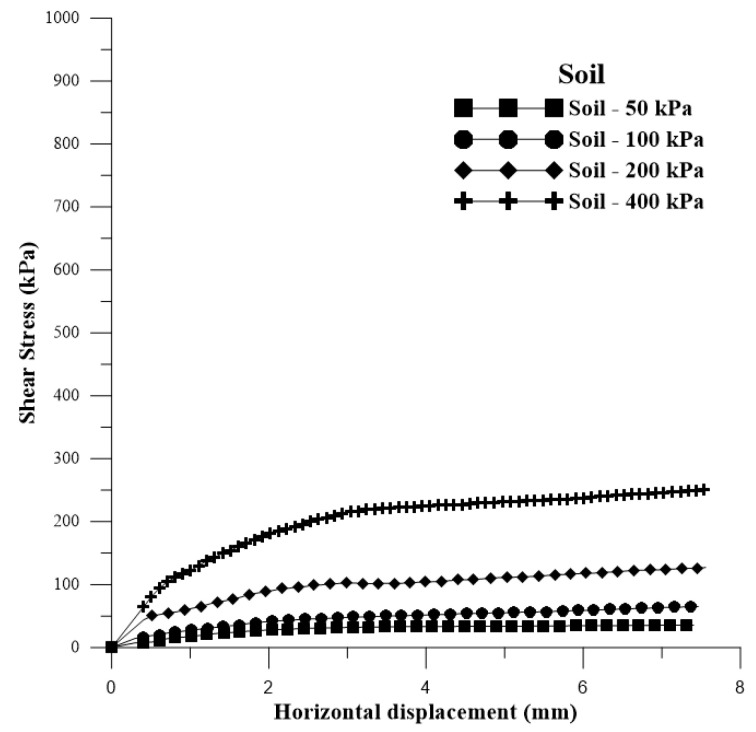
Shear stress versus horizontal displacement curves for soil + water.

**Figure 11 polymers-15-04626-f011:**
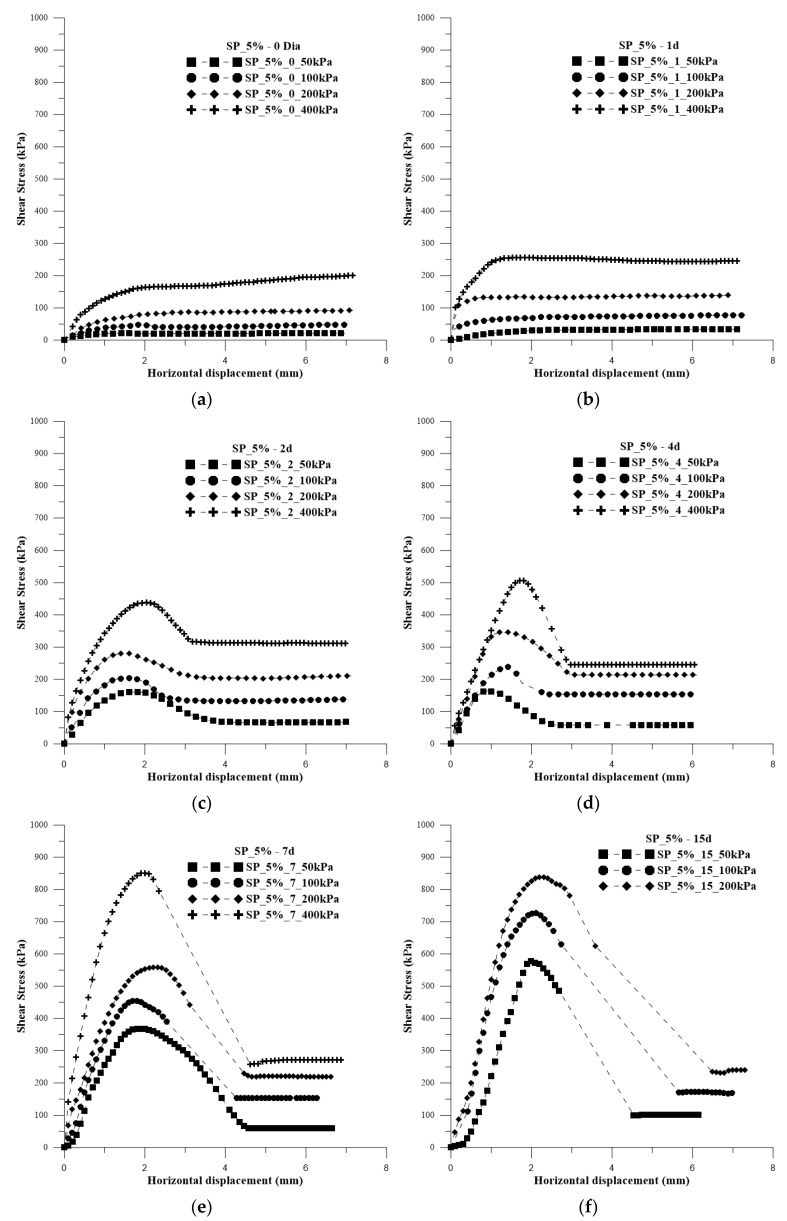
Shear stress versus horizontal deformation curves for each curing time and for the application of normal stresses of 50 kPa, 100 kPa, 200 kPa, and 400 kPa for 5% polymer: (**a**) 0 days of curing; (**b**) 1 day of curing; (**c**) 2 days of curing; (**d**) 4 days of curing; (**e**) 7 days of curing; (**f**) 15 days of curing; (**g**) 30 days of curing; (**h**) 45 days of curing.

**Figure 12 polymers-15-04626-f012:**
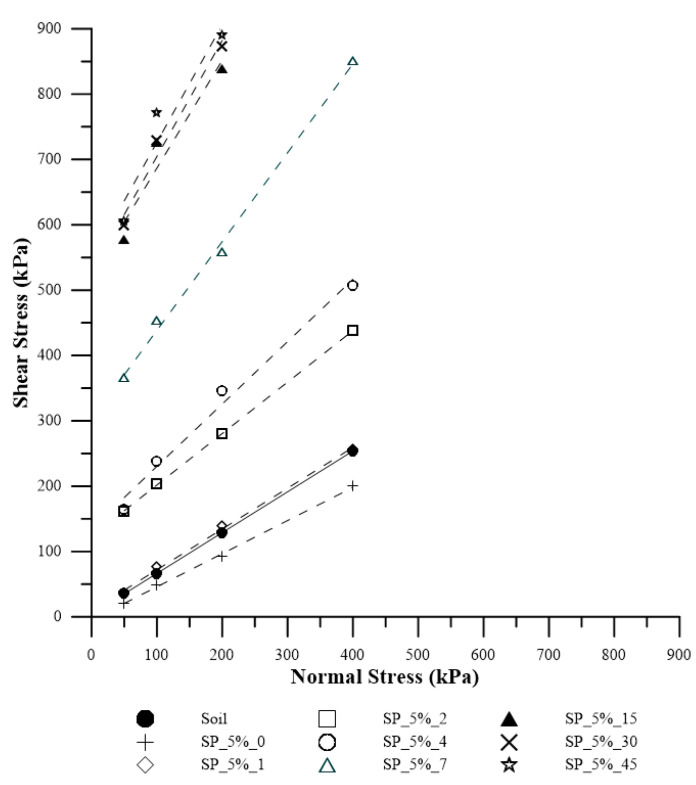
Rupture Envelopes for Soil and Composite SP_5% for Each Studied Curing Time.

**Figure 13 polymers-15-04626-f013:**
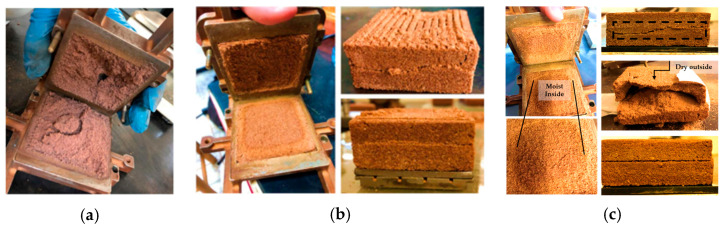
Specimens with 5% polymer after rupture in the direct shear test: (**a**) 0 days of curing; (**b**) 1 day of curing; (**c**) 2 days of curing.

**Figure 14 polymers-15-04626-f014:**
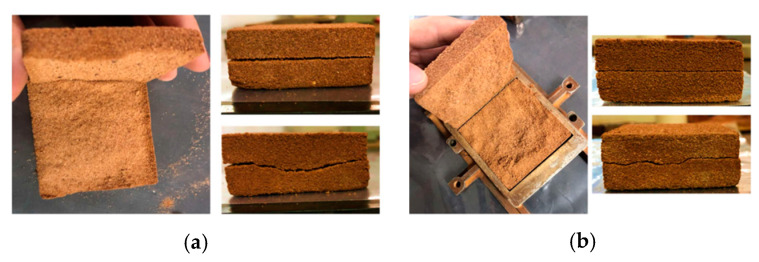
Specimens with 5% polymer after rupture in the direct shear test: (**a**) 15 days of curing; (**b**) 45 days of curing.

**Figure 15 polymers-15-04626-f015:**
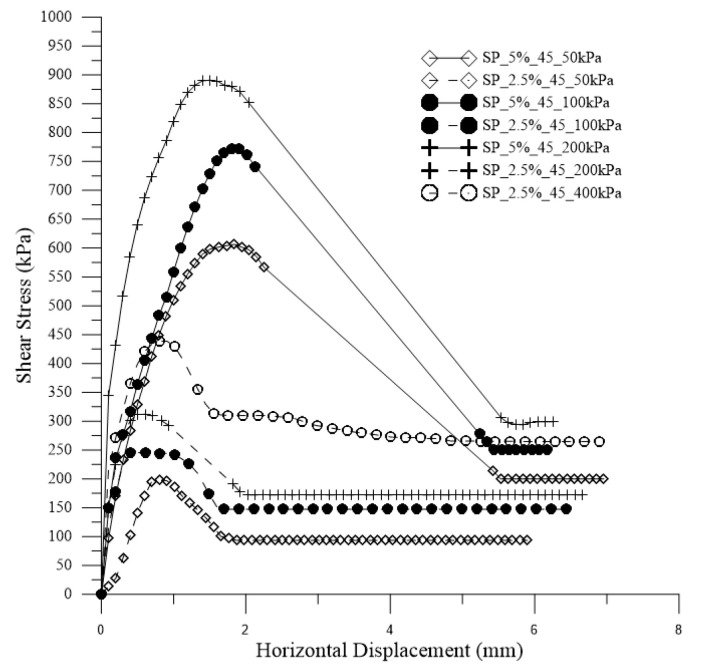
Stress vs. horizontal displacement curves for both composites with 45 days of curing and for the soil.

**Figure 16 polymers-15-04626-f016:**
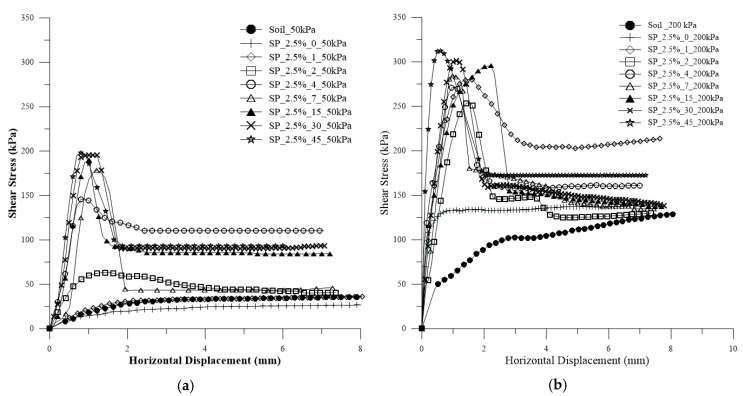
Curves for confining pressure for the SP_2.5% composite: (**a**) 50 kPa; (**b**) 200 kPa.

**Figure 17 polymers-15-04626-f017:**
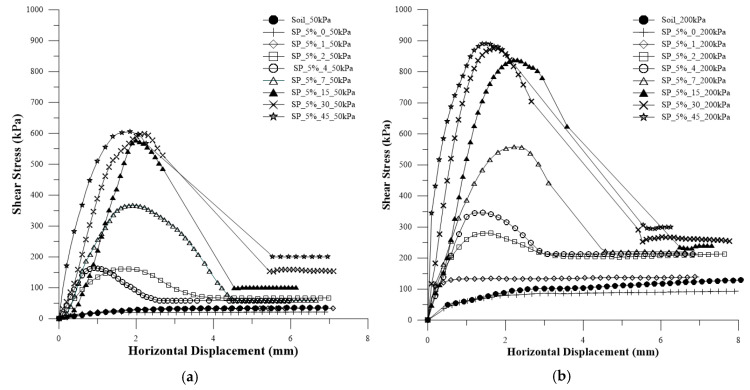
Curves for confining pressure for the SP_5% composite: (**a**) 50 kPa; (**b**) 200 kPa.

**Figure 18 polymers-15-04626-f018:**
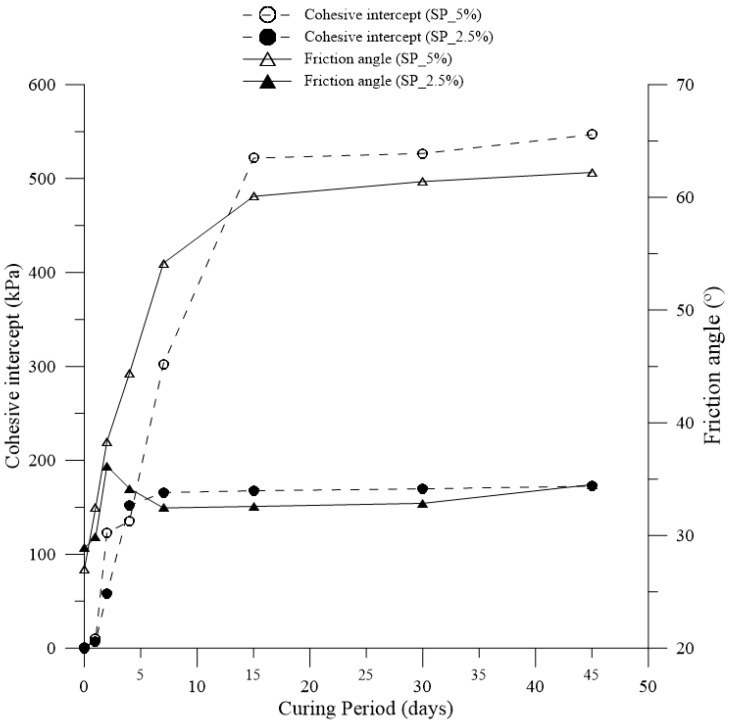
The correlation between the cohesive intercept and the friction angle of the composites and the corresponding curing time.

**Figure 19 polymers-15-04626-f019:**
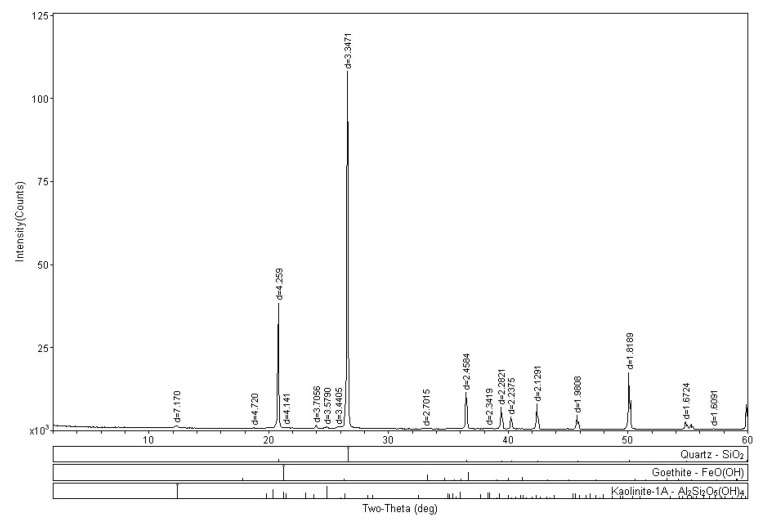
X-ray diffraction (XRD) test of sand.

**Figure 20 polymers-15-04626-f020:**
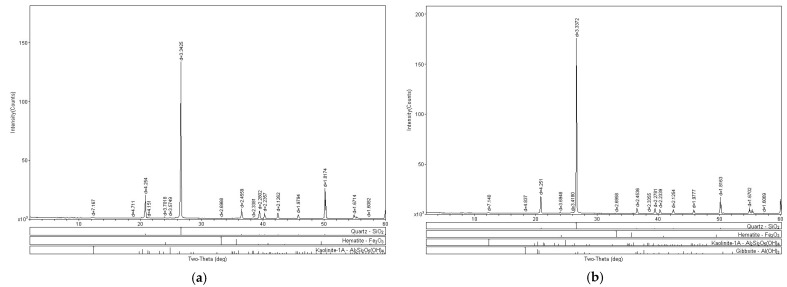
X-ray diffraction (XRD) tests: (**a**) SP_2.5% (**b**) SP_5%.

**Figure 21 polymers-15-04626-f021:**
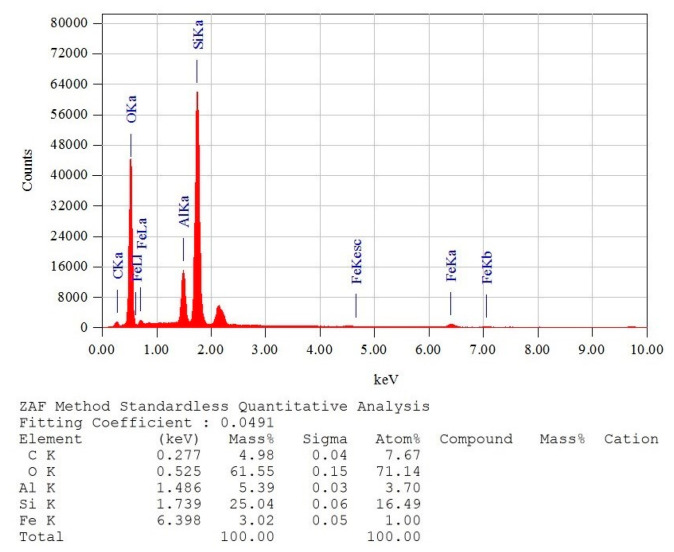
Energy Dispersive X-ray Spectroscopy (XRF/EDX) analysis of the soil.

**Figure 22 polymers-15-04626-f022:**
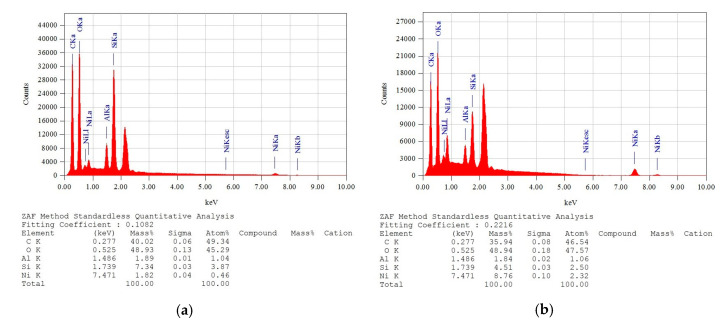
Energy Dispersive X-ray Spectroscopy (XRF/EDX) analyses: (**a**) SP_2.5% (**b**) SP_5%.

**Figure 23 polymers-15-04626-f023:**
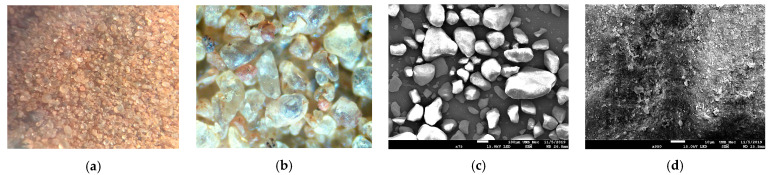
Images related to the studied soil: (**a**) Image obtained from Optical Microscopy (OM) at 4× magnification; (**b**) Image obtained from OM at 10× magnification; (**c**) Image obtained by Scanning Electron Microscopy (SEM) at 75× magnification; (**d**) Image of a soil grain obtained by SEM at 950× magnification.

**Figure 24 polymers-15-04626-f024:**
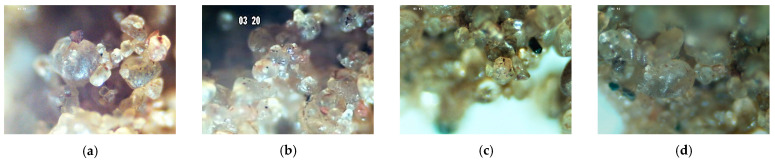
Optical Microscopy (OM) at 10× magnification: (**a**) External part of the SP_2.5% composite (**b**) Internal part of the SP_2.5% composite; (**c**) External part of the SP_5% composite (**d**) Internal part of the SP_5% composite.

**Figure 25 polymers-15-04626-f025:**
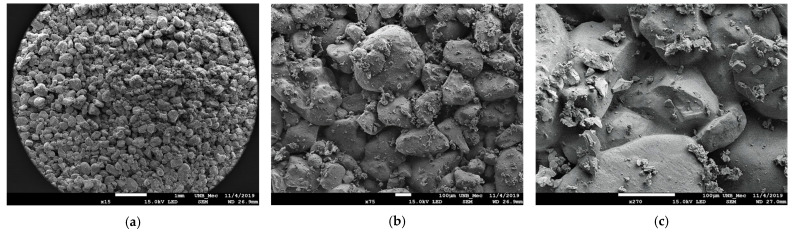
SEM images (SP_2.5%): (**a**) Magnification 15×; (**b**) Magnification 75×; (**c**) Magnification 270×.

**Figure 26 polymers-15-04626-f026:**
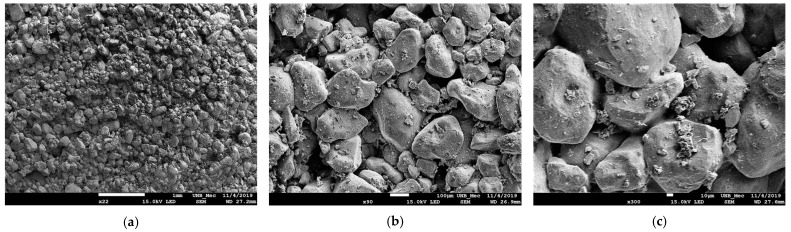
SEM images (SP_5%): (**a**) Magnification 22×; (**b**) Magnification 90×; (**c**) Magnification 300×.

**Table 1 polymers-15-04626-t001:** Values of effective friction angle and cohesive intercept for each curing time, SP_2.5%.

Material	c’ (kPa)	Φ’ (°)
Soil	4.05	31.9
SP_2.5%_0	1.06	28.9
SP_2.5%_1	7.37	29.8
SP_2.5%_2	57.81	36.2
SP_2.5%_4	151.95	34.2
SP_2.5%_7	165.70	32.4
SP_2.5%_15	167.59	32.6
SP_2.5%_30	169.67	32.9
SP_2.5%_45	172.53	34.5

**Table 2 polymers-15-04626-t002:** Values of effective friction angle and cohesive intercept for each curing time, SP_5%.

Material	c’ (kPa)	Φ’ (°)
Soil	4.05	31.9
SP_5%_0	0.00	27.0
SP_5%_1	9.68	32.5
SP_5%_2	123.12	38.3
SP_5%_4	134.96	44.4
SP_5%_7	302.48	54.2
SP_5%_15	521.86	60.1
SP_5%_30	526.72	61.4
SP_5%_45	546.72	62.2

**Table 3 polymers-15-04626-t003:** Chemical analysis conducted on the leachate at both dosages.

Chemical Elements	SP_2.5%	SP_5%
pH	8.8	9.0
Total Nitrogen (mg.L^−1^)	2.39	3.09
Iron (mg∙L^−1^)	<0.1004	<0.1004
Manganese (mg∙L^−1^)	<0.1004	<0.1004
Copper (mg∙L^−1^)	<0.1000	0.1091
Aluminum (mg∙L^−1^)	0.6110	0.6660

## Data Availability

Data are contained within the article.

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
