# Peer review of "The Application of an Eco-Friendly Synthetic Polymer as a Sandy Soil Stabilizer"

_polymers, 2023, doi:10.3390/polym15244626_

Round 1
Reviewer 1 Report
Comments and Suggestions for Authors
This manuscript details a study on the properties of a soil, from a protected region in Brazil, mixed with a polymeric solution. Shear testing was the most detailed characterization approach, with XRF, XRD and microscopy included also. While the paper is presented well and shear test results sufficiently described, there are a number of things to be addressed in the other areas of the manuscript.
1. There is little detail in the introduction about previous results from literature. The authors should also mention what differentiates this manuscript from others that have done similar work.
2. Details about the polymer used in this study are lacking. Just describing it as a "acrylic-styrene" copolymer is not sufficient in a polymers based journal. Things such as linear vs. branched and molecular weight distribution may have significant effects on the results but no such details were mentioned.
3. Section 2.4 should be expanded to include details of each experimental method and instrument used.
4. XRF vs. EDX typically describe different instruments but the results from both seem to be blended together throughout the manuscript. Was it a standalone XRF instrument or an EDX detector on the SEM or both? This needs to be explained.
4. Lines 125 - 131 should be grouped with the rest of the X-ray based discussion in Section 3.3.
5. In Line 322 the authors describe an increase in the friction angle on the fourth day, but Fig. 18 shows the peak on the second day.
6. In Section 3.3, the authors are attributing supposed hematite and gibbsite peaks in the XRD scan to the presence of the polymer in the composite. I fail to see how that's possible since there was no detail about the polymer structure to verify that claim. Also, there was iron in the bare soil. An XRD of the pure soil needs to be shown to see if these minerals were present prior to addition of the polymer. What was the percent match for those minerals due to the size of the peaks identified? A zoomed in version of the XRD scans and details of the scan parameters (see comment 3 above) would help here.
7. Control experiments of pure soil in XRD, SEM and OM results should be presented.
8. XRF/EDX compositional data should be displayed in a table to make it easier to compare.
9. Nickel is identified in XRF/EDX. While it can be present as a residue from the polymer synthesis process, the percentages reported here are extremely high. There should be an explanation for this given the small polymer fraction.
Comments on the Quality of English LanguageEnglish is generally fine. The authors should review the manuscript, especially the introduction, for typos and grammatical errors.
Author Response
The authors appreciate the comments provided by the reviewer, which have contributed to enhancing the content of the revised manuscript submitted herewith. To facilitate the review of the revised manuscript, the responses have been itemized in the attached document.

Reviewer 2 Report
Comments and Suggestions for Authors
The article should be revised according to the below comments.
1- Many grammatical and spelling mistakes are present and should be corrected.
2- The choice of the synthetic polymer should be justified.
3- The concentrations of the added polymer are not justified.
4- What about the dispersion state of the polymer?
5- What is the mechanism of stabilization and are there other polymers with similar ability?
6- The use of bio polymer may be preferred and should be at least discussed in the document. Further check the rules of green chemistry.
Comments on the Quality of English Language1- Many grammatical and spelling mistakes are present and should be corrected.
Author Response

(The authors gave the same response as above.)

Reviewer 3 Report
Comments and Suggestions for Authors
Dear Authors
I would like to congratulate you on the effort and work put into the analysis and preparation of the publication. The issue discussed is extremely interesting and involves significant challenges.
Due to personal interests, I would like to ask about one of the presented conclusions (lines 444 and 445)
From an environmental perspective, the leachate analysis did not reveal excessive chemical elements that could contaminate groundwater, fauna, flora, or the local population.
The above text does not specify what elements were obtained in the wastewater and whether, apart from the elements, any chemical compounds were specified?
Author Response

(The authors gave the same response as above.)

Round 2
Reviewer 1 Report
Comments and Suggestions for Authors
The manuscript has been improved.